# Bayesian Decision Analysis: An Underutilized Tool in Veterinary Medicine

**DOI:** 10.3390/ani12233414

**Published:** 2022-12-04

**Authors:** Charles O. Cummings, Mark A. Mitchell, Sean M. Perry, Nicholas Fleissner, Jörg Mayer, Angela M. Lennox, Cathy A. Johnson-Delaney

**Affiliations:** 1Tufts Clinical and Translational Science Institute, Tufts Medical Center, Boston, MA 02111, USA; 2Department of Veterinary Clinical Sciences, School of Veterinary Medicine, Louisiana State University, Baton Rouge, LA 70803, USA; 3Mississippi Aquarium, Gulfport, MS 39501, USA; 4Department of Surgical Sciences, School of Veterinary Medicine, University of Wisconsin-Madison, Madison, WI 53706, USA; 5Department of Small Animal Medicine and Surgery, College of Veterinary Medicine, University of Georgia, Athens, GA 30602, USA; 6Avian and Exotic Animal Clinic, Indianapolis, IN 46268, USA; 7Exotic DVM Forum, Edmonds, WA 98026, USA

**Keywords:** Bayes’ theorem, Bayesian, decision analysis, decision tree, clinical decision making, ferret, foreign body, cognitive bias

## Abstract

**Simple Summary:**

Decision making in veterinary medicine can be extremely difficult. Often, different choices can have vastly different costs, complications, and outcomes associated with them. Bayesian inference and decision analysis are two tools that, when combined, can help clinicians and pet owners decide on the preferred course of action. In this retrospective case study, we describe a lethargic ferret that is no longer eating. We solicited opinions from three expert veterinarians who were not involved with the case on what the diagnosis could be before and after a series of diagnostic tests. We also asked the original clinical team to estimate how valuable different clinical outcomes were. By combining these data, we were able to assess if the original clinical team was right to take the animal to surgery. We also discuss some of the pitfalls of not using Bayesian inference in diagnosis, some cognitive biases that may have played a role in the case management decisions, and the wider usefulness of decision-analysis methods to help foster shared decision making between client and veterinarian.

**Abstract:**

Bayesian inference and decision analysis can be used to identify the most probable differential diagnosis and use those probabilities to identify the best choice of diagnostic or treatment among several alternatives. In this retrospective case analysis, we surveyed three experts on the prior probability of several differential diagnoses, given the signalment and history of a ferret presenting for lethargy and anorexia, and the conditional probability of different clinical findings (physical, bloodwork, imaging, etc.), given a diagnosis. Using these data and utility estimates provided by other clinicians, we constructed a decision tree to retrospectively identify the optimal treatment choice between exploratory laparotomy and medical management. We identified medical management as the optimal choice, in contrast to the original clinical team which performed an exploratory laparotomy. We discuss the potential cognitive biases of the original clinical team. We also discuss the strengths, e.g., shared decision making, and limitations of a Bayesian decision analysis in the veterinary clinic. Bayesian decision analysis can be a useful tool for retrospective case analysis and prospective decision making, especially for deciding on invasive interventions or end-of-life care. The dissimilarity of expert-derived probability estimates makes Bayesian decision analysis somewhat challenging to apply, particularly in wide-ranging specialties like zoological medicine.

## 1. Introduction

A 75-year-old woman on warfarin therapy for chronic atrial fibrillation presented to the emergency room with gross hematuria [1]. Her prothrombin time was markedly prolonged. She was imaged with several different modalities which found two soft tissue masses in one kidney. She had two urine cytologies reviewed, but both were negative for neoplastic cells [1]. All imaging was considered to be consistent with potential renal neoplasia [1]. She underwent a radical nephrectomy; however, on histopathology there was no evidence of neoplasia [1]. In fact, when another physician reviewed the patient’s chart and analyzed the decisions made, he found that the pre-surgery probability of this patient having renal cancer was less than or equal to 1 in 10 million [2]. How is this possible?

It seems that a lack of probabilistic thinking was to blame. The physicians involved in the case did not consider the probability of renal carcinoma vs. over anticoagulation as the cause of hematuria in an elderly woman on warfarin. The clinicians involved in this case expressed concerns about what the phrase “considered to be consistent with renal cell cancer” in an imaging report actually meant [1]. Did inclusion of that phrase mean there was a 90% likelihood of renal cell cancer, a 10% likelihood, or just that it could possibly be renal cell cancer?

Bayesian inference, based on Bayes’ theorem, is a way to incorporate additional information into prior hypotheses in order to improve clinical decision making. Bayes’ theorem states that the probability of a diagnosis given a set of clinical findings is proportional to the baseline probability, or prevalence, of that diagnosis multiplied by the probability of having that set of clinical findings given that diagnosis. Bayesian inference considers both how common a diagnosis is in the population and how well the clinical findings fit the diagnosis. This is important as uncommon presentations of common diseases can be far more prevalent than “textbook presentations” of rare diseases.

We seek here to introduce Bayesian decision analysis to a veterinary audience and apply it to a case example in an exotic companion mammal, a ferret (*Mustela putorius furo*).

## 2. The Principles of Bayesian Decision Analysis

In order to identify a most likely differential diagnosis, there must be some data or an estimate of the relative prevalence of the differential diagnoses (adding up to 100%) for a specific presentation in a given population; in the prior example, this would be the prevalence of renal cancer vs. bladder cancer vs. benign causes as potential causes of hematuria in an overly anticoagulated elderly woman. These prior probabilities are proposed before information from diagnostic tests are incorporated; prior probabilities can be based on literature estimates of prevalence and risk factors, expert opinion, or an experienced clinician’s understanding of how common different diagnoses are in their particular clinic. For each potential diagnosis, there is also a conditional probability associated with each diagnostic test which is the estimated probability of that specific test result given that the patient has a particular diagnosis (e.g., the probability of a negative urine cytology assuming the patient has renal cell cancer). A related but distinct value, the likelihood ratio, is the ratio of conditional probabilities for those with a particular test result and the disease, and those with the same result without the disease. By multiplying each diagnosis’ prior probability by its conditional probabilities, we can calculate posterior probabilities, which are the estimated probabilities of differential diagnoses after the information from diagnostic tests has been incorporated.

Decision analysis is a field of study that mathematically models decisions in order to identify the best choice among alternative options. In clinical medicine, decision analysis evaluates both risks and benefits associated with a treatment or diagnostic test [3]. To model a clinical decision, clinicians estimate the utility, or value, of clinical outcomes in addition to diagnosis probabilities as above. Utility can be expressed as the survival benefit, with 100 representing 100% survival and 0 representing 0% survival, but often represents a composite figure, termed relative utility, of both survival and costs (medical and financial) incurred [3]. By multiplying the posterior probabilities by the utility of each outcome, clinicians can better identify the treatment option with the greatest expected value for their patient [3].

## 3. Materials and Methods

To illustrate the application and utility of Bayesian decision analysis, a veterinary clinical case was retrospectively analyzed. In that case, the primary decision hinged on whether to perform exploratory laparotomy for a suspected gastrointestinal foreign body. First, the clinical case was briefly described. Second, outside expert opinions were elicited as to the probability of a gastrointestinal foreign body vs. six non-surgical diagnoses. Third, utility estimates were elicited from the original clinical team for each of four potential clinical outcomes: surgery + gastrointestinal foreign body (GI FB), surgery + no GI FB, no surgery + GI FB, no surgery + no GI FB. Finally, a decision tree was made to identify the optimal decision using the utility estimates and diagnosis probabilities.

### 3.1. Case Description

An approximately 1-year-old, castrated male ferret presented emergently with a 2-day history of lethargy and anorexia. One day prior to presentation, the patient developed diarrhea. At presentation, the owners mentioned that he was a “chewer.” Abnormal physical exam findings included a rectal temperature of 40.6 °C (105.1 °F; normal values: 37.8–40.0 °C [100–104 °F]) [4] and a mildly enlarged left axillary lymph node. Three view radiographs indicated the stomach was moderately distended with inhomogeneous soft tissue opaque material (Appendix A). An ultrasound evaluation by a non-radiologist found gastric foreign material but no other abnormalities. A complete blood count (CBC) and plasma biochemistry panel were also performed (Appendix A).

The patient was taken to surgery for exploratory laparotomy. The stomach was found to be full of food material with no foreign material identified. There was some erythematous gastric mucosa identified and biopsied, but no other abnormalities were found within the abdomen.

The patient had an extremely prolonged recovery after surgery (several hours). Two days post-surgery, the patient was still extremely lethargic, anorexic, and had a fever of 40.1 °C (104 °F). Despite the risk of exacerbating potential GI ulceration, a high dose of meloxicam (0.5 mg/kg) was administered to reduce the fever. The next day, however, the patient’s fever had increased to 40.6 °C (105.1 °F). Given the lack of response to therapy, the owners elected to humanely euthanize the patient three days after the initial presentation. Submission of the gastric biopsy and necropsy were declined by the owner. An axillary lymph node fine needle aspirate submitted shortly after surgery revealed reactive lymphoid hyperplasia. For an expanded case description, radiographic, CBC, and other laboratory data, see the Appendix A.

### 3.2. Case Analysis

#### 3.2.1. Diagnosis Probability Estimates

In order to avoid hindsight and outcome bias [5], three clinicians with significant expertise in ferret medicine, unassociated with the case, were consulted (J.M., A.M.L., C.A.J.-D.). Using an online survey, clinicians were given the patient’s signalment and history and were asked to assign prior probability estimates for each of the following seven differential diagnoses (deemed most likely diagnoses by the first author) as primary causes of the patient’s clinical signs: lymphoma, gastrointestinal foreign body (GI FB), systemic coronaviral infection, *Helicobacter* gastritis, disseminated idiopathic myofasciitis (DIM), bacterial gastroenteritis, and unknown toxicosis.

Clinicians were given the diagnostic findings—physical exam, radiography, ultrasonography, plasma chemistry panel, and complete blood count—one at a time, using the clinical information and images as written and shown in an extended case description (Appendix A). Clinicians were reminded to only consider each diagnostic finding independently. To estimate the conditional probabilities, these clinicians were asked to answer the question, “What percentage of patients with diagnosis X would have similar findings?” for each diagnostic finding-diagnosis combination. Estimates could range from 100, meaning all patients with disease X would have similar findings, to 1, meaning ≤ 1% of patients with disease X would have similar findings. This minimum estimate of 1% was set because lower estimates, like 1 in 1000 or 10,000, were thought unlikely to be precise, given that even experienced ferret clinicians may have seen far fewer than 100 similar cases.

Composite prior and conditional probabilities were calculated by averaging clinician estimates [6]. Posterior probabilities of pre-surgical findings were calculated for each diagnosis (Table 1). Clinicians were also asked to provide conditional probabilities for the surgical findings, results of lymph node cytology, and the clinical progression post-surgery given each differential diagnosis. The composite pre-surgical posterior probabilities, characterized here as prior probabilities, were multiplied by these additional composite-conditional probabilities to calculate the posterior probabilities of differential diagnoses at the time of euthanasia (Table 2). Iterative calculations of posterior probabilities were also performed after each diagnostic, pre- and post-surgery, to illustrate how differential diagnosis probabilities changed as new evidence was incorporated into the clinical picture (Figure 1).

For each diagnosis, a posterior probability was calculated using the following equation:$$\mathrm{Posterior}\;\mathrm{Probability}\;(\%)=\frac{\mathrm{Prior}\;\mathrm{Probability}\;\times \;\mathrm{Conditional}\;{\mathrm{Probability}}_{1}\;\times \;\mathrm{Conditional}\;\mathrm{Probabilityn}\ldots }{\mathrm{Sum}\;\mathrm{of}\;\mathrm{All}\;\mathrm{Diagnoses}’\;\mathrm{Prior}\;\times \;\mathrm{Conditional}\;\mathrm{Probabiliies}}\;\times \;100$$

**Table 2 animals-12-03414-t002:** Outside clinician-derived composite prior (posterior probabilities from Table 1 become prior probabilities due to a different reference time point) and conditional probabilities for differential diagnoses and diagnostic findings after all pre-surgical diagnostics in an approximately 1-year-old ferret presenting for two days of lethargy and inappetence, and one day history of diarrhea. Posterior probabilities of differential diagnoses at time of euthanasia are shown.

Diagnosis	Prior Probability (%)	Conditional Probability (%) Exploratory Laparotomy	Conditional Probability (%) Lymph Node Cytology	Conditional Probability (%) Clinical Progression	Prior × Conditional Probabilities	Posterior Probability (%)
Lymphoma	5	23.7	35	15	62,801	10
GI Foreign Body	31	12	6.7	11.7	28,658	4
Systemic Coronaviral Infection	0.1	7	11.7	13.3	145	0.02
*Helicobacter* Gastritis	5	60	6.1	19.3	36,966	6
Disseminated Idiopathic Myofasciitis	6	12	6.1	55	26,146	4
Bacterial Gastroenteritis	52	51.7	6.1	30.7	501,131	76
Unknown Toxicosis	1	8.7	5.3	13.3	350	0.05
SUM					656,197	

For each diagnosis, a posterior probability was calculated using the following equation:$$\mathrm{Posterior}\;\mathrm{Probability}\;(\%)=\frac{\mathrm{Prior}\;\mathrm{Probability}\;\times \;\mathrm{Conditional}\;{\mathrm{Probability}}_{1}\;\times \;\mathrm{Conditional}\;\mathrm{Probabilityn}\ldots }{\mathrm{Sum}\;\mathrm{of}\;\mathrm{All}\;\mathrm{Diagnoses}’\;\mathrm{Prior}\;\times \;\mathrm{Conditional}\;\mathrm{Probabiliies}}\;\times \;100$$

#### 3.2.2. Outcome Utility Estimates

There are several different ways to elicit utility estimates, including rating scales, standard gambles, and time tradeoffs, all of which are described in greater depth elsewhere [6]. In this case, we elected to use a rating scale. Specifically, the clinicians who initially saw the patient (C.O.C., M.A.M., S.M.P.) were asked to estimate the relative utility (a composite figure incorporating both survival and cost) of three clinical outcomes on a scale of 0–100: surgery + GI FB, surgery + no GI FB, no surgery + GI FB (Table 3). The fourth clinical outcome of no surgery + no GI FB was assigned the maximum utility of 100, because it meant that surgery was not required and not performed, sparing both money and potential adverse effects from an unnecessary procedure. Clinicians were explicitly instructed to consider how the cost of an unnecessary surgery would affect utility.

Treatment thresholds were calculated using the utility estimates in the following equation:$$\frac{1}{1+[((\mathrm{Treated}\;\mathrm{disease}\;-\;\mathrm{Untreated}\;\mathrm{disease})\;/\;(\mathrm{Untreated}\;\mathrm{no}\;\mathrm{disease}\;-\;\mathrm{Treated}\;\mathrm{no}\;\mathrm{disease}))]}\;=\;\mathrm{Treatment}\;\mathrm{threshold}.$$

For the composite utilities, the calculations were performed as follows: $$\frac{1}{1+[((83.3\;-\;23.3)\;/\;(100\;-\;60))]}=0.4\;\mathrm{or}\;40\%$$

#### 3.2.3. Decision Tree & Treatment Threshold Sensitivity Analysis

A decision tree was constructed with each clinical outcome as a terminal (rightmost) branch (Figure 1). The utility of each outcome was then multiplied by its associated diagnosis probability to provide an expected value of each option, with the higher value considered to be the optimal choice (in this case surgery vs. no surgery).

To understand the lowest diagnostic probability at which surgery should be elected (treatment threshold), a one-way sensitivity analysis was performed [7]. For this, both the benefits and harm associated with surgery were calculated. The benefit of surgery was the difference in utility between the surgery + GI FB (treated disease) and no surgery + GI FB outcomes (untreated disease). The harm of surgery was the difference in utility between the no surgery + no GI FB (untreated no disease) and surgery + no GI FB (treated no disease) outcomes. The treatment threshold is calculated with the equation: $\frac{1}{1+(\mathrm{Benefit}/\mathrm{Harm})}$ [7]. In this model, a treatment or diagnostic test with a high benefit:harm ratio (low risk, low cost) should be selected, even with a relatively low confidence of the diagnosis. Alternatively, treatments or diagnostic tests with a low benefit:harm ratio (high risk, high cost) require a much higher degree of certainty regarding the diagnosis. An overview of the approach to clinical decision analysis using the “PROACTIVE” mnemonic is presented in Appendix A [6].

## 4. Results

### 4.1. Diagnosis Probability Estimates

Each outside clinician noted gastrointestinal foreign body as the differential diagnosis with the highest prior probability. However, these probabilities varied widely, with one clinician estimating a 30% prior probability and another doubling that estimate at 60% (Appendix A). Prior probabilities for other differentials were generally concordant, although one clinician considered *Helicobacter* gastritis 10 times more likely and bacterial gastroenteritis 2.5 times more likely than another. Estimated conditional probabilities of both pre-surgical and post-surgical findings were often concordant, but in some cases showed marked differences (Appendix A).

Composite pre-surgery posterior probability estimates noted bacterial gastroenteritis (52%) followed by GI foreign body (31%) as the most likely differential diagnoses (Table 1). Individual pre-surgery posterior probability estimates from each clinician differed dramatically from one another, and the clinicians indicated *Helicobacter* gastritis (59%), lymphoma (92%), and bacterial gastroenteritis (93%) to be the most likely pre-surgery differential diagnosis (Appendix A). While composite pre-surgery posterior probability estimates indicated a 31% chance of a GI foreign body, no clinician had an individual pre-surgery posterior probability estimate of a GI foreign body above 7%. At the time of euthanasia, the individual estimates of posterior probabilities again varied widely (Appendix A). Estimates for two of the three clinicians indicated that, at the time of euthanasia, lymphoma was the most likely diagnosis (86 and 99.1%), and bacterial gastroenteritis was the second least likely diagnosis (0.4 and 0.0000009%). Estimates for the third clinician indicated that, at the time of euthanasia, bacterial gastroenteritis was the most likely differential diagnosis (99.7%) and lymphoma was the third least likely diagnosis (0.000004%). Composite posterior probabilities of diagnoses at the time of euthanasia indicated that bacterial gastroenteritis was the most likely diagnosis at 76%, while gastrointestinal foreign body was tied for the fourth most likely diagnosis at 4% (Table 2).

### 4.2. Outcome Utility Estimates

While no surgery + no GI FB (untreated no disease) was assigned the highest utility, the three clinicians on the original clinical team agreed on the relative order of the other outcomes (Table 3). The outcome with the second highest composite utility was surgery + GI FB (treated disease, utility of 83.3/100), followed by surgery + no GI FB (treated no disease, utility of 60/100), then no surgery + GI FB (untreated disease, utility of 23.3/100) as the worst outcome. Utility estimates varied somewhat by clinician (Table 3).

### 4.3. Decision Tree & Treatment Threshold

The decision tree (Figure 2) revealed no surgery to be preferred to surgery given that the composite pre-surgery posterior probability of GI FB was 31%. A one-way sensitivity analysis to determine the treatment threshold showed that surgery would have been preferred to no surgery if the probability of a foreign body was above 40%. A probability of exactly 40% would indicate neither strategy is preferable to the other. This treatment threshold varied slightly when using individual clinician estimates of utility (Table 3).

## 5. Discussion

The primary aim of this investigation was to introduce Bayesian decision analysis and apply it to a clinical case, specifically to answer the question, “Should this patient have undergone surgery?” It is an important question. Unnecessary surgery has profound costs, including medical costs in the form of adverse anesthetic or surgical events, as well as financial opportunity costs, (e.g., money that could have been spent on other diagnostics or medical management). Based on probability estimates from clinicians not involved in the case and utility estimates from clinicians who were, the answer to that question was “No, this animal should not have gone to surgery as there was only a 31% chance that it had a GI foreign body, and the treatment threshold was 40%.”

While knowledge gaps, communication errors, and circumstance (e.g., Friday night presentation) may have played a role in this suboptimal decision, it is always important to consider the cognitive biases that contributed to it as well. Doing so may help to mitigate them in the future. Multiple related biases—premature closure, anchoring, and confirmation bias—may have been at play in this case. In premature closure bias, there is a failure to adequately consider alternative diagnoses after an initial clinical impression is formed. In this case, the initial clinical impression based on signalment and history was consistent with a GI foreign body, and many other diagnoses were not explicitly considered during initial discussions amongst the team. Anchoring bias is the overreliance on earlier information or failing to update one’s prior probabilities in light of information that is not supportive of those probabilities. In this case, despite the physical exam and radiographic findings that outside clinicians thought would be somewhat unusual (20%) in ferrets with GI foreign bodies, the original clinicians still considered it to be the most likely diagnosis. To confirm this, they performed an abdominal ultrasound to find gastric foreign material. This was an example of confirmation bias, namely seeking or preferring test results to confirm rather than disproving a suspected diagnosis.

To illustrate how these biases would alter pre-surgery diagnosis posterior probabilities, suppose that the original clinicians considered the diagnostic prior probabilities and the conditional probabilities for ultrasonographic findings (confirmation bias), but discounted all other pre-surgical diagnostics (anchoring bias). In that scenario, the pre-surgery posterior probability of GI foreign body was 64% compared to 31% when all pre-surgical diagnostics were considered. In light of these cognitive biases, it becomes obvious why the original clinicians elected to take the patient to surgery.

One strength of this decision-analysis approach is that it can allow for improved, shared decision making between client and veterinarian, especially when it comes to deciding on potentially invasive or expensive therapies/diagnostics. To derive utilities for use in the decision tree, clinicians can explain the benefits and harms of a treatment or test to the client, whether it be surgery, renal biopsy, radiation therapy, or a drug with potential adverse effects. Clients, on the other hand, will have a better understanding of how much they value each outcome financially. Together, clinicians and clients can assess the potential outcomes and assign utility values to them. This can also be done when there are more than two options from which to choose. For example, in the case presented, another option could have been endoscopic evaluation and the removal of a gastric foreign body. Because of this, decision analysis allows clinicians to help their clients determine the best option for them between the most efficacious and often most expensive therapy (the gold standard) and other less costly but less efficacious alternative therapies.

Another strength of this approach is that because decision analysis makes the decision-making process explicit, clinicians are able to assess the robustness of their decisions to different diagnosis probabilities or utility values. The one-way sensitivity analysis performed above can determine the treatment threshold, but this can also be extended to identify a test-treatment threshold, which shows the probabilities at which a diagnosis should be treated without additional testing, tested for, or not treated without additional testing [8]. Although different clinicians may disagree about the structure of the problem (e.g., the number of alternatives), diagnosis probabilities, utilities, and thus treatment thresholds, decisions made using decision analysis are defensible as rational.

One major limitation of decision analysis is that clinicians do not necessarily have the time to implement it. Clinicians make dozens of decisions every day for which decision analysis would be excessive. Rather, clinicians could view decision analysis as a tool to facilitate shared decision making for end-of-life care or when deciding whether to go forward with particularly invasive diagnostics or interventions like surgery, mechanical ventilation, or hemodialysis. Another limitation of Bayesian decision analysis, especially in this zoological medicine context, was the imprecision and dissimilarity of the probability estimates (Appendix A). This is inescapable given the relative lack of relevant published data. Even in human medicine, probability estimates vary in degrees of fuzziness, depending on the specialty and clinical scenario. Accordingly, some have even defined clinical decision analysis as “a formal approach to making bad decisions, which would have otherwise been worse” (J. Wong, personal communication). Some of this imprecision in literature-derived estimates can stem from the use of language such as, “frequently,” “commonly,” “uncommonly,” or “rarely” when describing clinical signs of a disease as opposed to data-driven percentage estimates. Clinician-derived estimate imprecision is also likely to be common in exotic animal/zoological medicine, a field defined by its breadth rather than depth. Compared to the high number of heart attacks and kidney stones that an emergency physician may see, zoological medicine clinicians encounter diseases and species almost at random. We attempted to account for this stochasticity of experience by surveying more than one experienced clinician. Furthermore, limitations of data imprecision apply to the entirety of exotic animal/zoological practice, so this does not preclude clinicians from using the Bayesian decision analysis to understand and improve their own clinical decision making. Readers can use a fill-in-the-blank template for their consideration in using Bayesian inference for diagnosis (Figure 3, Appendix A).

## 6. Conclusions

Clinical decision making is difficult. Retrospectively, humans are plagued with hindsight and outcome biases such that when we misdiagnose patients or have bad outcomes, we come to believe that the right diagnosis was more obvious or that the treatment choice made was clearly wrong. Prospectively, there are a myriad of unknowns that complicate the decision-making process. In this article, we have described the Bayesian decision analysis as a tool for ordering differential diagnoses, making prospective clinical decisions, and analyzing past clinical decisions (Appendix A). We have also applied this tool in a retrospective case analysis of a ferret with a suspected gastrointestinal foreign body. Given the exact same information, outside clinicians came to different conclusions about prior diagnosis probabilities and conditional probabilities of clinical findings. By combining these expert opinions into composite estimates, we endeavored to reflect a collective truth rather than individual experiences. In doing so, we were able to analyze our own decisions and understand the cognitive biases that may have given rise to them. We hope others are able to do the same and start improving their own clinical reasoning.

## Figures and Tables

**Figure 1 animals-12-03414-f001:**
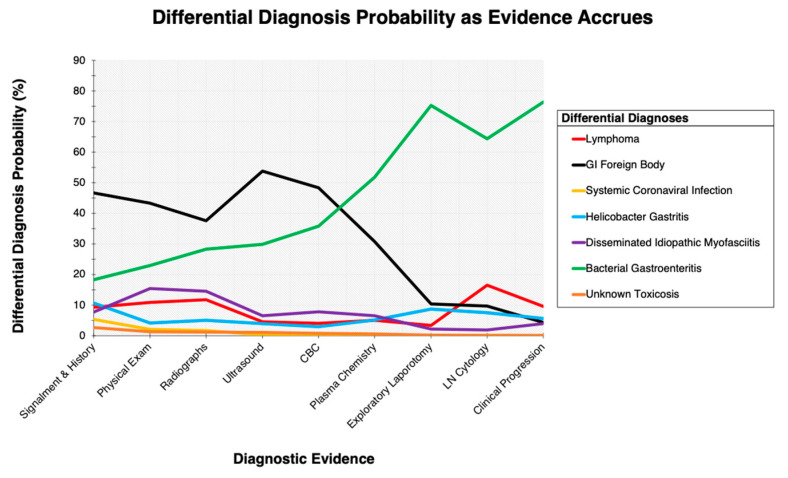
Each of seven differential diagnosis probabilities for a 1-year-old castrated male ferret that presented to a veterinary teaching hospital following two days of lethargy and inappetence, and one day of diarrhea. The cumulative composite posterior probabilities for each differential diagnosis change as each new piece of diagnostic evidence is incorporated into the clinical picture.

**Figure 2 animals-12-03414-f002:**
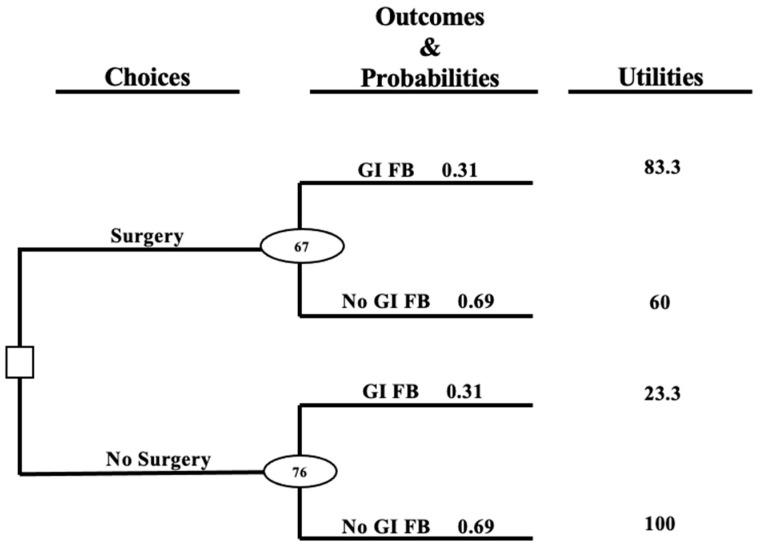
Decision tree for the clinical case of a ferret with a possible gastrointestinal foreign body (GI FB). The square node represents a decision to be made by the clinician, while the oval nodes, called chance nodes, represent the different outcomes dictated by diagnosis probability. The numbers in the oval nodes represent expected values (∑ (utility × diagnosis probability)) of each choice. For example, the composite probability of a GI FB at time of surgery (0.31) was multiplied by the utility value of the surgery + GI FB outcome (83.3) to equal 25.8; then the probability of not having a GI FB (1 − 0.31 = 0.69) was multiplied by the utility value of the surgery + no GI FB outcome (60) to equal 41.4. When 41.4 and 25.8 were added together and rounded to the nearest whole number, the expected value of surgery was revealed to be 67. This process was repeated for the “no surgery” choice to reveal an expected value of 76. In this case, the “no surgery” choice provides greater value.

**Figure 3 animals-12-03414-f003:**
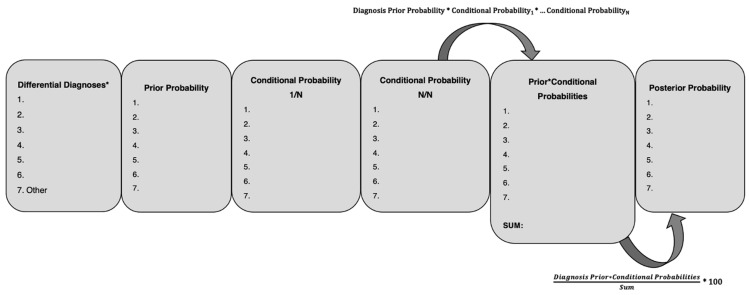
Template for using Bayes’ theorem to think probabilistically in diagnosis. Differential diagnoses should be exhaustive and mutually exclusive, meaning they should cover the realm of possibilities and that only one is the true primary cause of disease. If necessary, inclusion of an “other” category, consisting of aggregated low probability diagnoses, may be considered. Conditional probabilities may come from literature, expert opinion, or in the case of certain diagnostic assays, data from the manufacturer or lab. Any probability, prior or conditional, of 0 will result in that diagnosis having a posterior probability of 0, so it is important to consider whether a diagnosis or a diagnostic finding is possible, i.e., has a probability > 0, rather than simply unlikely. Posterior probability in this template is expressed as a percentage, rather than on a 0 to 1 scale. An interactive spreadsheet formulation has been included in the Appendix A to allow easier application of this tool.

**Table 1 animals-12-03414-t001:** Outside clinician-derived composite prior and conditional probabilities for differential diagnoses and diagnostic findings, respectively, in an approximately 1-year-old ferret presenting for two days of lethargy and inappetence, and one day history of diarrhea. Pre-surgical posterior probabilities are also shown.

Diagnosis	Prior Probability (%)	Conditional Probability (%) Physical Exam	Conditional Probability (%) Radiography	Conditional Probability (%) Ultrasonography	Conditional Probability (%) CBC	Conditional Probability (%) Chemistry	Prior × Conditional Probabilities	Posterior Probability (%)
Lymphoma	9.3	25	25	11.7	35	30	71,458,333	5
GI Foreign Body	46.7	20	20	43.3	35	15.3	434,103,704	31
Systemic Coronaviral Infection	5.3	8.3	18.3	5.3	23.7	18.3	1,885,542	0.1
*Helicobacter* Gastritis	10.7	8.3	28.3	23.3	30	41.7	73,456,790	5
Disseminated Idiopathic Myofasciitis	7.7	43.3	21.7	13.7	46.7	20	91,816,379	6
Bacterial Gastroenteritis	18.3	27	28.3	32	46.7	35	733,040,000	52
Unknown Toxicosis	2.7	10.3	21.7	26.7	27	18.7	8,024,178	1
SUM							1,413,784,925	

**Table 3 animals-12-03414-t003:** Individual and composite (average) estimates of relative utility for different clinical outcomes from the original clinical team. The utility for no surgery + no gastrointestinal foreign body was assumed to be 100 (maximum).

Clinical Outcome	Individual Estimated Utility			Composite Estimated Utility
Surgery + Gastrointestinal Foreign Body	90	80	80	83.3
Surgery + No Gastrointestinal Foreign Body	70	60	50	60
No Surgery + Gastrointestinal Foreign Body	25	25	20	23.3
No Surgery + No Gastrointestinal Foreign Body	100			
Treatment Threshold	32% chance of GI FB	42% chance of GI FB	45% chance of GI FB	40% chance of GI FB

## Data Availability

Not applicable.

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
