# Peer review of "Bayesian Decision Analysis: An Underutilized Tool in Veterinary Medicine"

_animals, 2022, doi:10.3390/ani12233414_

Round 1

Reviewer 1 Report

This is a very interesting manuscript written in an excellent manner investigating if the Bayesian Decision Analysis which can be used for decision-making under uncertainty can be used to improve veterinary decision-making in clinical practice. Using a case study they describe the process, the benefits but also the limitations this technique has are clearly described. The paper is very easy to read and follow. 

The paper among other elements describes how this could benefit decision-making e.g. to avoid cognitive and anchoring biases, to better inform clients and to make a more robust decision. The paper also clearly describes the limitations such as first and foremost the lack of reliable data, and the time restraint. More case studies to use it, e.g. to facilitate shared decision-making for end-of-life care would be interesting.

From the many papers I reviewed, it is one of the best-written papers, so I have very few comments. 

My comments: 

- one of the big limitations (also recognised by the paper) is the limited data available in veterinary medicine e.g. on the prevalence of certain diseases in certain species. If we look at how these data could be collected; then this is the easiest to do by veterinary corporates having many veterinary practices if they use the same data management system with the same coding (which is often not the case at this moment). For independent veterinary practices; this is much more difficult.  

- Line 348 Readers can use a fill-in-the-blank template for their 348 consideration in using Bayesian inference for diagnosis (Figure 3). Even better would be to have an electronic collection and calculation system. 

- typo description Table 1? This Outside clinician-derived []

Reviewer 2 Report

Review of manuscript animals-2003530

I enjoyed reading your paper. I found the manuscript novel and, in theory, understood the idea. As a clinician interested in minimizing diagnostic errors, this paper added a new method to help in this area. Calculating a diagnostic utility value is very important for clinicians as we regularly have clients with limited funds, and we want to do the best for the pet and the client.

My major problem with the paper is understanding how you derived many values. I am not good at statistics, but I do have some understanding. Because I want to try your technique with selected cases, I went through the Tables and Figures in the paper and Supplemental Material. I understood about 60%. I had problems with how some values were derived, and my questions focus on these issues.

I think I am a good “guinea pig” for testing the practical application of your methods. I especially like your template in Fig 3. However, I don’t see a Template for calculating the Utility. I had problems with Table S4.

I am not making “must-do” recommendations. Instead, I am raising the areas where I had problems. I would like you to think of how you can explain better to non-mathematically oriented veterinary practitioners so they can use your template and come up with utility values.

I recommend “revisions” and look forward to reading your revised paper. I am enthusiastic about the publication of your article.

1 In Table 2 (and Table S3), should not the second column heading labeled as “Prior Probability” be “Posterior Probability?” The value you have in the last column of Table 1 for a lymphoma diagnosis is 5 (in column Posterior Probability (%)). So, in Table 2, the value in the 2nd column labeled “Prior Probability (%)” for lymphoma is 5. That’s the value from Table 1. So, shouldn’t the Prior Probability column in Table 2 be labeled Posterior Probability?

2 Fig 1. Are the diagnostic probabilities cumulative? i.e., using GI foreign body as an example. The diagnostic probability is ~47% after signalment and history. It is ~ 38% when radiographs are used. Is the 38% at radiographs a cumulation of (signalment & history + physical exam + radiographs)? Or is the 38% an independent diagnostic probability based only on radiographs? 

3 Fig 2. I don’t understand where the values of 67 and 76 in the oval nodes came from. I also don’t understand where all the values next to Surgery  GI FB 0.31, No GI FB 0.69, and No surgery GI FB 0.31, No GI FB 0.69, come from. 

4 Table S4. I can see where the Utilities values of 83.3, 60, 23.3, and 100 came from. But I don’t understand where each Individual Estimated Utility value came from. In your text (lines 178-180) you have written, “the clinicians … were then asked to estimate the relative utility (a composite figure incorporating both survival and cost) of the three remaining clinical outcomes: surgery + GI FB, surgery + no GI FB, no surgery + GI FB (S6).” The supplemental material does not have a S6. 

5 Lines 237-239. A one-way sensitivity analysis to determine the treatment threshold showed that surgery would have been preferred to no surgery if the probability of a foreign body was above 40% percent.” Can you show the one-way sensitivity analysis in Supplemental Material?

Round 2

Reviewer 2 Report

I recommend publication of your article.

Thank you for making changes to your manuscript that allowed me to understand every step. I tried a hypothetical case (including your Excel sheet interactive calculator) and felt confident on your Excel sheet that I duplicated what you are doing. I will try your Bayesian Decision Analysis with our residents next time we have a group case discussion meeting.